Identification of keratinocyte-associated genes for immune characterization and drug response prediction in oral squamous cell carcinoma

Huang Jinyu 1
Li Yijing 2
Lin Weiyi 3
Wu Zhiqin 4
Si Shanshan sishanshan@smu.edu.cn 5
Shu Dalong shudl@mail.sysu.edu.cn 3
1 Department of Endodontics, Stomatological Hospital, School of Stomatology, Southern Medical University , Guangzhou , China
2 Department of Orthodontics, Stomatological Hospital, School of Stomatology, Southern Medical University , Guangzhou , China
3 Department of Stomatology, The First Affiliated Hospital of Sun Yat-sen University , Guangzhou , China
4 Department of Stomatology, Jiangxi University of Chinese Medicine , Nanchang , China
5 Department of Oral Emergency, Stomatological Hospital, School of Stomatology, Southern Medical University , Guangzhou , China
Guan Fanglin
Electronic publication date: 2025 Oct 28
Publication date: 2025
Volume: 13
Electronic Location ID: e19953
Received 2025 May 14; Accepted 2025 Jul 29
Copyright: ©2025 Huang et al.
Copyright year: 2025
Copyright holder: Huang et al.
License: This is an open access article distributed under the terms of the Creative Commons Attribution License, which permits unrestricted use, distribution, reproduction and adaptation in any medium and for any purpose provided that it is properly attributed. For attribution, the original author(s), title, publication source (PeerJ) and either DOI or URL of the article must be cited.
License URL: https://creativecommons.org/licenses/by/4.0/

Keywords: High-dimensional weighted correlation network analysis, Oral squamous cell carcinoma, Keratinocyte, Immune characterization, Biomarkers

Funding: The authors received no funding for this work.

==============================
Background

Oral squamous cell carcinoma (OSCC) is one of the most frequent types of head and neck tumor. Keratinocytes play a crucial part in tumor cell growth but their role in OSCC remains unknown.

Methods

We obtained single-cell RNA sequencing (scRNA-seq) data and bulk RNA sequencing data of OSCC from the Gene Expression Omnibus (GEO) database and utilized the Seurat package for quality control, downscaling, and clustering of the scRNA-seq data. The CellChat package was utilized to develop a ligand-receptor network of keratinocytes. Subsequently, high-dimensional weighted gene co-expression network analysis (hdWGCNA) and differential expression analysis were employed to identify keratinocyte-related gene modules and obtain hub genes. The predictive value of the hub genes was assessed by constructing a diagnostic model, and the CIBERSORT and ESTIMATE algorithms were utilized to analyze the correlation between immune infiltration and the diagnostic model. Finally, the mRNA expressions of the screened genes were measured, and their effects on the proliferation, migration, and invasion ability of OSCC cells were explored using in vitro models.

Results

We identified eight major cellular subpopulations including T cells and keratinocytes. Cellular communication revealed that keratinocytes may have close mutual communication with macrophages, fibroblasts, and endothelial cells. The hdWGCNA screening classified nine keratinocyte-related modules and 50 hub genes were extracted, among them KRT6B, KRT16, CSTB, and CSTA were identified as differentially expressed keratinocyte-related genes. A nomogram was developed, and KRT16, CSTA, and CSTB were determined as highly effective genes for the diagnosis of OSCC. Immune infiltration analysis revealed that StromalScore, ImmuneScore and ESTIMATEScore, were negatively linked to CSTA and CSTB but positively correlated with KRT16. Finally, in vitro experiments showed that the viability, migration, and invasion of OSCC cells were markedly suppressed after knockdown of KRT16.

Conclusion

Our study provided novel biomarkers targeting keratinocytes for the treatment of OSCC.

Introduction

Oral squamous cell carcinoma (OSCC) is a frequently detected malignancy in the oral and maxillofacial region (Mu et al., 2023; Ru & Zheng, 2024; Zhang et al., 2024). The 5-year overall survival (OS) rate for OSCC patients remains below 50% in the past 20 years (Warnakulasuriya, 2009; Panzarella et al., 2014). Study showed that a poor prognosis of OSCC is largely caused by the aggressive metastatic potential of the cancer, which encourages OSCC cells to move to distant organs such as the lungs (SHahinas & Hysi, 2018; Dong, Zhang & Chang, 2023). Though current therapeutic strategies, for instance, surgery, radiation therapy, and chemotherapy are the main treatments for OSCC, patients still face unfavorable survival and a high risk of recurrence (Scully & Bagan, 2009). While recent studies have provided several diagnostic markers for OSCC (Qi & Tang, 2024; Yue & Yao, 2023; Liu, Wang & Li, 2022), the identification of more effective markers is still of critical importance for the diagnosis and prognostic improvement of OSCC.

Abnormal keratinization, nuclear division, and cell multinucleation are morphological abnormalities that enable the malignant transformation of epithelial cells during the development of OSCC (Foki et al., 2020; Luo et al., 2018). Typical keratinization process of keratinocytes is different from these aberrant alterations. For example, OSCC is characterized by poor keratinization, heterogeneous proliferation, large nuclei, and increased chromatin. These pathohistological features can be observed under a microscope because cancer cells show irregular shapes, size variations, and abnormal nucleoplasm proportion (Liu et al., 2024). A study has reported that aberrant keratinocyte proliferation and differentiation are closely linked to OSCC progression (Nguyen et al., 2022). Abnormal proliferation and differentiation of cells not only alter the shape and function of the cells, but also confer tumor cells with malignant features. Thus, discovering keratinocyte biomarkers in OSCC is crucial for understanding the tumor pathophysiology, development of specific treatment plans, and prognostic prediction. The technique of single-cell RNA sequencing (scRNA-seq) has allowed researchers to clarify tumor complexity and heterogeneity at the single-cell level (Kurten et al., 2021), in particularly, OSCC has been widely studied with scRNA-seq analysis (Wang et al., 2024). The intricate interactions between stromal cells, epithelial cells, and immune cells in the tumor microenvironment (TME) can also be thoroughly investigated by the scRNA-seq analysis (Li et al., 2023; Wang et al., 2021b). Moreover, scRNA-seq analysis also contribute to developmental biology by enabling comprehensive characterization of cellular heterogeneity and their corresponding gene expression.

In this research, we defined distinct keratinocyte subpopulations in OSCC using scRNA-seq data and performed comprehensive characterization of keratinocytes by computational analysis. Our goal was to reveal the molecular features of keratinocytes in OSCC, identify potential prognostic markers and therapeutic targets for OSCC, hoping to improve the clinical treatment of OSCC.

Methods

Data sources

The Gene Expression Omnibus (GEO; https://www.ncbi.nlm.nih.gov/geo/) dataset was accessed to obtain the scRNA-seq data, clinical information, and survival information of 12 OSCC samples (GSE215403). In addition, the dataset of OSCC microarray data (GSE30784) was also downloaded and screened to obtain bulk RNA-seq data of 45 normal oral tissue samples and 167 OSCC tissue samples.

Processing and analysis of the scRNA-seq data of OSCC

Quality control and cell filtering of the scRNA-seq data of OSCC were conducted using the “Seurat” package (Butler et al., 2018). A total of 31,776 cells were retained after eliminating those with 200 to 6,000 expressed genes and more than 10% mitochondrial content. After normalizing and scaling the raw data using the SCTransform function, principal component analysis (PCA) was performed (Bahudian & Valdovinos, 2025). The “harmony” package (Korsunsky et al., 2019) was employed to remove the batch effect amongst various samples, followed by conducting uniform manifold approximation and projection (UMAP) for dimensionality reduction using the RunUMAP function. Next, we built a K nearest neighbors (KNN) network based on the Euclidean distance with the top 50 principal components employing the FindNeighbors function. The cells were then clustered into subpopulations by the FindCluster function at the resolution of 0.05. The intercellular communication of keratinocyte subpopulations mediated by ligand–receptor pairs was analyzed using the R package “CellChat”(Efremova et al., 2020), with “Cell-Cell Contact” as the interaction type. Finally, the results were visualized into bubble plots.

Analysis of specific high-expressed genes and functional annotation between cellular subpopulations

The FindAllMarkers tool (parameters: “logfc.threshold = 0.30, min.pct = 0.30, only.pos = T”) was applied to screen specifically high-expressed genes (significance threshold of p < 0.05) from various subpopulations as cellular markers. Functional annotation of differentially expressed genes (DEGs) was performed using the package “CusterProfiler” (Yu et al., 2012).

High-dimensional weighted gene co-expression network analysis screening of keratinocyte-related genes

According to the characteristics in scRNA-seq data, high-dimensional weighted gene co-expression network analysis (hdWGCNA) can be employed to explore particular gene expression patterns involved in various biological processes (Langfelder & Horvath, 2008) and build co-expression networks across several cellular and spatial hierarchical scales (Morabito et al., 2023). Here, we used the hdWGCNA to read single-cell transcriptome rds data to develop the co-expression network of keratinocytes under the optimal soft threshold of eight. The modules related to keratinocytes were sectioned by calculating the connectivity of the modules, and the key genes within each module were selected based on their connectivity to the modules for further study.

Identification of DEGs

DEGs between the normal and OSCC tissue samples in the GSE30784 dataset were filtered using the “limma” package (Ritchie et al., 2015) under the criteria of |log2FC| ≥ 1 and padj < 0.01. Then, genes present in intersection between the selected modular genes and the DEGs were considered as differentially expressed keratinocyte-related hub genes.

Construction of a gene diagnostic model

We first tested the diagnostic potential of the hub genes using the R package “multipleROC” (Cook, 2008). Subsequently, the R package “rms” was used to construct a diagnostic model for predicting the risk of OSCC, and a nomogram was drawn (Iasonos et al., 2008). The discriminative performance of the model was compared based on the receiver operating characteristic (ROC) and area under the ROC curve (AUC). The accuracy and efficiency of the model were validated by plotting calibration curves with the “caret” package, and decision curve analysis (DCA) (Gerds, Andersen & Kattan, 2014) was conducted utilizing the “rmda” R package.

Immune infiltration analysis

The official CIBERSORT website (https://cibersortx.stanford.edu/) was accessed to source the expression data of 22 types of common immune infiltrating cells (LM22), and the level of immune cell infiltration between OSCC samples and control samples in the GSE30784 dataset was calculated using the “CIBERSORT” R package (Newman et al., 2015). Rank-sum test was applied to determine significant immune infiltration differences (p < 0.05) between the two types of samples, while Spearman’s correlation coefficient (p < 0.05) was used to calculate the relationship between immune cell infiltration and crucial genes. Using the R language’s “estimate” package (Yoshihara et al., 2013), immune infiltration in the GSE30784 dataset was examined and the results were displayed as corresponding scores. The correlation between the pivotal genes and ESTIMATEScore, StromalScore, and ImmuneScore was analyzed according to the Spearman’s correlation coefficient, and the rank-sum test was employed to determine whether the immune scores were significantly different between the two types of samples (p < 0.05).

Cell culture and plasmid transfection

Human normal oral keratinocytes (HOK; CP2610) were ordered from ScienCell Research Laboratories (Carlsbad, CA, USA). Human OSCC cells (HSC-3; JCRB0623) were purchased from the Japanese Collection of Research Bioresources (JCRB, Osaka, Japan). Human oral squamous carcinoma cells (WSU-HN30; STM-CL-5612) were purchased from Stemmer Biotechnology Co (Shanghai, China). DMEM medium (Gibco, Thermo Fisher Scientific Inc., Waltham, MA, USA) supplemented with 1% penicillin-streptomycin (15140122; Gibco) and 10% fetal bovine serum (FBS; 10099141C, Gibco) was utilized for cell culture in an incubator with 5% CO2 at 37 °C. The cells were identified by Short Tandem Repeat (STR) analysis, and the results of mycoplasma detection for these cells were negative.

According to the instruction of Lipo3000 Liposome Transfection Reagent (L3000-001; Thermo Fisher Scientific Inc.), WSU-HN30 and HSC-3 cells (2 × 104 cells/well) in the logarithmic growth phase were transfected with KRT16 knockdown plasmid (si-KRT16, sequence: 5’AACAGCGAACTGGTACAGAGC 3’) and control plasmid (si-NC), which were purchased from GenePharma (Shanghai, China).

RNA extraction and quantitative real-time polymerase chain reaction

Total RNA was extracted from HOK, WSU-HN30, and HSC-3 cells using the RNA Extraction Kit (TRIzol, 15596026; Invitrogen, Carlsbad, CA, USA), following the protocols. The purity and concentration of the total RNA were determined, and cDNA template was generated using the HiScript II First-strand cDNA Synthesis kit (R211-01; Vazyme, Nanjing, China) (Zhang et al., 2023). The quantitative real-time polymerase chain reaction (qRT-PCR) was conducted using specific primers and the KAPA SYBR® FAST kit (KK4600; Sigma Aldrich, Burlington, MA, USA). The data were calculated using the 2−ΔΔCT method with GAPDH as the internal control. See Table 1 for the primer sequences of the specific genes.

Table 1 Primer sequences applied in this study.

Gene	Forward primer	Revers primer	
KRT16	5′CTACCTGAGGAAGAACCACGAG 3′	5′CTCGTACTGGTCACGCATCTCA 3′	
CSTA	5′AAACTCAAGTTGTTGCTGGAACAAA 3′	5′TTTGTCAACCTGGTATCCAGTAAG 3′	
CSTB	5′CGTGTCATTCAAGAGCCAGGTG 3′	5′GCTTGGCTTTGTTGGTCTGGTAG 3′	
GAPDH	5′GTCTCCTCTGACTTCAACAGCG 3′	5′ACCACCCTGTTGCTGTAGCCAA 3′	

Cell viability

According to the protocol, CCK-8 (CK04; Dojindo, Tokyo, Japan) was performed to assess the effect of KRT16 on the viability of WSU-HN30 and HSC-3 cells. Briefly, the cells (5 × 103 cells/well) were grown in 96-well microtiter plates for 24, 48, and 72 hours (h). After washing the cells twice with phosphate-buffered saline (PBS), 100 µL of fresh medium and CCK-8 solution (10 µL) was added to each well for 3-h incubation at 37 °C with 5% CO2. Absorbance (at 450 nm) was measured in a SPECTROstar® Nano (BMG LABTECH GmbH, Ortenberg, Germany) (Ma et al., 2023).

Cell migration assay

For wound healing assay, the transfected cells were inoculated into 6-well plates (5 × 105 cells/well) and 2 ml of cell suspension was added for incubation in an incubator at 37 °C with 5% CO2. When the cells adhered to the wall, the monolayer was wounded with a 10 µL plastic pipette tip to form a uniform scratch. PBS was used for washing the monolayers, which were then incubated in medium without FBS. The wound edge distances between the two edges of the migrating cell sheet were photographed at 0 h and 48 h. All the experiments were conducted in triplicate.

Cell invasion assay

WSU-HN30 and HSC-3 cell suspensions (5 × 105 cells/well) were prepared in serum-free medium. Then, 100 µL of cell suspension was supplemented to the upper Transwell chamber (Corning, Corning, NY, USA) pre-coated with Martigel (30 µg/well; BD, San Jose, CA, USA), while the lower chamber contained 600 µL of DMEM medium and 10% FBS. The migrated cells were fixed by 4% paraformaldehyde and colored by crystal violet solution. The cells in the lower chamber were counted under a microscope from six different fields of view.

Cell apoptosis assay

For the flow cytometry assay, the cells were rinsed in phosphate buffered saline and suspended using 1× binding buffer provided by the assay kit (C1062S, Beyotime, China) to adjust to the concentration of 1 × 106 cells/mL. 100 µL cell suspension was then taken and added with the working solution of Annexin V-FITC and propidium iodide (5 µL) for a 10-minute mixture reaction in the dark at ambient temperature. The apoptosis was finally tested in a flow cytometer (BD, USA).

Statistical analysis

GraphPad Prism 8 (GraphPad Software, San Diego, CA, USA) and R software version 3.6.0 (R Foundation, Vienna, Austria) were utilized in all statistical analyses. Data were presented as mean ± standard deviation and analyzed using the Student t-test or two-way ANOVA followed by Bonferroni correction for post hoc multiple comparisons. A p < 0.05 stood for statistically significant difference.

Result

Identification of cell subtypes in OSCC

We obtained a total of 12 cell clusters after processing the scRNA-seq data of OSCC using the “Seurat” package (Fig. 1A), and eight major cell subpopulations were identified, namely, T cells, keratinocytes, macrophages, B cells, fibroblasts, plasma cells, endothelial cells, and mast cells (Fig. 1B). Subsequently, cell type annotation was performed using the CellMarker2.0 database to identify representative markers for T cells (CCL5, GNLY, and GZMA), keratinocytes (KRT17, KRT14, and S100A8), macrophages (LYZ, C1QB, and C1QA), B cells (CD37, CD79A, and MS4A1), fibroblasts (DCN and LUM), plasma cells (IGHG1 and IGHG4), endothelial cells (VWF and PECAM1), and mast cells (TPSB2 and CPA3). The normalized expression levels of these marker genes were displayed as bubble plots (Fig. 1C). Keratinization of the oral mucosal epithelium has been reported to affect the immune response, and immune checkpoint-associated factors are low-expressed in OSCC of keratinized epithelial origin (Kitsukawa et al., 2024). To further explore the potential impact of keratinocytes on OSCC, we performed enrichment analysis on specifically high-expressed genes of the keratinocyte subpopulation. It was found that these genes were significantly enriched in establishment of protein localization to endoplasmic reticulum entries, translational initiation, signal-recognition particle (SRP)-dependent cotranslational protein targeting to membrane, and endoplasmic reticulum-targeted protein (Fig. 1D, p < 0.05).

Figure 1 Annotation and identification of key cell subtypes in OSCC.

(A) Distribution of different samples after de-batching. (B) UMAP visualization of the distribution of different cell types. (C) Expression levels of different cell marker genes. (D) Functional enrichment analysis of keratinocyte-specific highly expressed genes.

Cellular communication between keratinocytes and other cells in OSCC

The reciprocal receptor–ligand pair communication network between keratinocytes and other cell subtypes in OSCC was explored employing CellChat analysis, which showed that keratinocytes communicated with other cellular subtypes. The ligand–receptor information for each cluster was extracted, and we discovered that endothelial cells affected keratinocytes through SELE-CD44 and APP-CD74, fibroblasts affected keratinocytes through APP-CD74, and macrophages affected keratinocytes through HLA-DRB5-CD4 (Fig. 2). These results indicated that keratinocytes and cell subpopulations had a high level of reciprocal communication.

Figure 2 Bubble plot demonstrating the ligand-receptor pairs of cellular regulation of keratinocyte.

Keratinocyte-related modules screened by hdWGCNA

To develop a co-expression network, a soft threshold power of β = 8 was found to achieve an optimal network connectivity when the scale-free topology fit reached 0.90 (Fig. 3A). Additionally, nine modules (Keratinocyte-M1 to Keratinocyte-M9) were identified (Fig. 3B) by calculating the expression of module eigengenes and module connectivity. The top 10 core genes of each module were selected based on the connectivity of the characterized genes (Fig. 3C).

Figure 3 HdWGCNA analysis of OSCC-associated keratinocytes.

(A) Soft threshold screening to obtain the optimal soft threshold value of 10. (B) In a co-expression network tree, the upper half is the gene tree, each branch refers to a gene, and the lower half is the module corresponding to the gene. (C) In module division, the vertical coordinate is the kME value, which represents the connectivity of each gene based on the feature gene, and the right side is the hub gene of the module.

Subsequently, we calculated the expression of genes in each module in different cells. Notably, the genes in the M3 and M5 modules showed high levels of activation mainly in keratinocytes (Fig. 4A), therefore these two modules were regarded as the key modules. Then, correlation analysis that M6 was negatively correlated with the M3 and M5, while strong positive correlations were observed between the M3, M4, M7, and M9 as well as between the M1, M6, and M8 (Fig. 4B). Then, co-expression network for the two modules were mapped using their top 25 hub genes (Figs. 4C–4D), with the 10 most highly connected genes in the inner circle as key genes and 15 genes in the outer circle as secondary key genes.

Figure 4 HdWGCNA marks the construction of a keratinocyte co-expression network associated with OSCC.

(A) Expression of genes within the module on different cells, red color represents high expression, blue color represents low expression, and the size of the circle indicates the proportion of cells. (B) Module correlation matrix. Where green color indicates a negative correlation, while purple color reflects a positive relation between gene modules. The circle size reflects the strength of the correlation. (C) Module Keratinocyte-M3 hub gene co-expression network. (D) In the module Keratinocyte-M5 hub gene co-expression network, the inner circle is the key gene and the outer circle is the minor key gene.

DEG analysis and screening of signature genes between OSCC and control samples

The “limma” package screened 1573 DEGs (789 upregulated genes and 784 downregulated genes) between OSCC and control samples (Fig. 5A). Next, the intersection of the hub genes of the two key modules and the DEGs contained four genes (KRT6B, KRT16, CSTB, and CSTA), which were considered as differentially expressed keratinocyte-related genes (Fig. 5B). The expression heatmap of the four genes showed that KRT6B and KRT16 were high-expressed in OSCC samples, while CSTB and CSTA genes were low-expressed (Fig. 5C). A diagnostic model for OSCC was developed using the four genes, and the model reliability was assessed according to the ROC curves and AUC. It was found that the AUC of KRT6B, KRT16, CSTB, and CSTA reached an AUC of 0.791, 0.917, 0.95, and 0.902, respectively (Fig. 5D), with KRT16, CSTA, and CSTB showing an AUC greater than 0.9. This indicated a strong diagnostic value of the three genes in OSCC, and these genes were employed in subsequent analysis.

Figure 5 Screening of OSCC-characterized genes.

(A) Volcano plot demonstrating the differentially expressed genes; gray represents non-differentially regulated genes, blue represents differentially downregulated genes, and red represents differentially upregulated genes. (B) Upset plot for differentially expressed genes as well as hdWGCNA module hub gene; the left bar is the number of each subset of genes, and the top bar is the number of each intersecting gene. (C) Heatmap of the expression of the four intersecting genes; blue color represents low expression and red color represents high expression. (D) ROC curves of the four genes.

Diagnostic model construction

To further improve the risk prediction for OSCC patients, we build a nomogram combining the expression of the three genes (Fig. 6A). The model showed that CSTB had the greatest impact on OSCC prognosis, followed by KRT16, while CSTA had limited impact on the prediction of the OS. Next, the ROC curve showed that the AUC of the model was 0.996 (Fig. 6B), implying that the diagnostic model was highly reliable. The calibration curve demonstrated that the calibration curve was close to the standard one, which suggested that the nomogram had strong prediction performance (Fig. 6C). In addition, the reliability of the model was tested utilizing the DCA decision curve. It can be observed that the gain of the nomogram was notably higher than the gain of other individual genes, suggesting a strong predictive ability of our model (Fig. 6D).

Figure 6 Nomogram construction for predicting the prognosis of OSCC patients.

(A) Nomogram with three biomarker expressions in the center the score corresponding to the expression at the top, and the risk probability at the bottom by calculating the total score ratio. (B) ROC curve of the nomogram. (C) The correction curve of the nomogram. (D) Decision curve of the nomogram.

Correlation of the three characterized genes with immune infiltration

Immune cell infiltration differences between OSCC samples and control samples in the GSE30784 dataset were compared. The results of CIBERSORT showed that the infiltration of most immune cell types was notably different between the two types of samples. In particular, cells such as macrophages M1 (M1), macrophages M0 (M0), and mast cells activated had higher levels of infiltration in OSCC patients than in control samples, while B cells memory, T cells CD8, immune cells such as macrophages M2 (M2), and mast cells resting had lower levels of infiltration (Fig. 7A). Subsequently, we calculated the correlation between the expression of the characterized genes and the immune cell infiltration scores. As shown in Fig. 7B, the expressions of the characterized genes were linked to the infiltration of a majority of immune cells. Specifically, the expression of CSTA and CSTB was positively linked to the infiltration scores of B cells memory, T cells CD8, M2, dendritic cells resting, and mast cells resting (p < 0.01) but negatively correlated (p < 0.001) with the infiltration of cells such as M0, M1, and mast cells activated. In contrast, the expression of KRT16 was positively associated with the infiltration scores of mast cells activated, M0, and M1 (p < 0.01) but negatively associated with the infiltration scores of B cells memory and T cells CD8 (p < 0.001). Immune microenvironment differences between OSCC samples and control samples were also compared. ESTIMATE analysis revealed that StromalScore, ImmuneScore, and ESTIMATEScore were all significantly higher in OSCC samples than in control samples (p < 0.001), indicating a higher level of immune cell infiltration in the OSCC samples (Fig. 7C). We also found that the expression of CSTA and CSTB was negatively correlated with StromalScore, ImmuneScore, and ESTIMATEScore in OSCC (p < 0.001), while KRT16 expression was positively linked to ESTIMATEScore and StromalScore (Fig. 7D, p < 0.05). These results indicated that the characterized genes may be closely associated with immune cell infiltration in OSCC.

Figure 7 Immunological characterization between OSCC samples and control samples.

(A) Box line plot of infiltration status of 22 types of immune cells CIBERSORT. (B) SPEARMAN correlation analysis of the expression of three characterized genes with 22 immune cell infiltration profiles. (C) ESTIMATE assessment of differences in immune infiltration scores between OSCC samples and control samples. (D) The Spearman correlation analysis of three characterized genes with immune infiltration status. ***, p < 0.001; **, p < 0.01; *, p < 0.05.

Downregulation of KRT16 suppressed the migratory and invasive abilities of OSCC cells

The mRNA expressions of three key genes in HOK, WSU-HN30 and HSC-3 cells were detected by performing qRT-PCR. The results demonstrated that the expression of KRT16 was significantly upregulated in WSU-HN30 and HSC-3 cells than in HOK cells, while the expression of CSTA and CSTB was significantly lower in WSU-HN30 and HSC-3 cells than in HOK cells (Fig. 8A). As we have confirmed a high diagnostic value of KRT16 and the potential role of its overexpression in the malignant progression of OSCC, we prioritized KRT16 for functional validation. After the knockdown of KRT16 gene in OSCC cells (Fig. 8B), it was observed that KRT16 knockdown significantly inhibited the survival of WSU-HN30 and HSC-3 cells (Fig. 8C). Subsequently, wound healing and transwell assays showed that KRT16 knockdown markedly suppressed the migration and metastasis of WSU-HN30 and HSC-3 cells (Figs. 8D–8G) but notably increased apoptosis in WSU-HN30 and HSC-3 cells (Fig. 8H). These results indicated a potential role of KRT16 in the occurrence of OSCC and its development.

Figure 8 Exploring the biological role of KRT16 in OSCC.

(A) The expression levels of KRT16, CSTA, and CSTB in HOK, WSU-HN30 and HSC-3 cells detected by qPCR. (B) Verification of the knockdown efficiency of KRT16 by qPCR in WSU-HN30 and HSC-3 cells. (C) Validation of the effect of knockdown of KRT16 on the viability of WSU-HN30 and HSC-3 cells. (D–G) Statistical analysis of representative images and invasive cell counts in wound healing assay and transwell assay of WSU-HN30 and HSC-3 cells after KRT16 knockdown. (H) The apoptosis level of WSU-HN30 and HSC-3 cells after KRT16 knockdown. The data are expressed as mean ± standard deviation, *, p < 0.05; **, p < 0.01; ***, p < 0.001; ****, p < 0.0001.

Discussion

OSCC is characterized by high aggressiveness, molecular heterogeneity, and varied treatment responses. Though recent research in the discovery of distinct tumor cell subpopulations has greatly improved our understanding of OSCC (Puram et al., 2017), the clinical significance of these subpopulations and the underlying mechanism of their behaviors in OSCC still remain mostly unknown. Emerging evidence has shown crucial functions of keratinocytes in OSCC (Foki et al., 2020; Hakelius et al., 2013). According to recent research, keratinocyte genes are linked to the development and spread of hepatocellular carcinoma (Zhao et al., 2022) and prostate cancer, showing the potential to serve as biomarkers for tumor prognosis (Liang et al., 2018). In addition, tumor-adjacent keratin-forming cells produce injury-related pro-inflammatory factors, contributing to the growth, migration, invasion, and metastasis in melanoma (Dainese-Marque et al., 2024). According to recent research, keratinocytes with high gene expressions are considered as risk factors for OSCC (Wang et al., 2021a). These findings suggested that keratinocytes may play a crucial part in the onset and spread of OSCC, and that identifying keratinocyte-associated indicators in OSCC may help understand the invasive mechanisms in OSCC.

This identified the keratinocyte subpopulation as a distinctive cellular subpopulation of OSCC based on the single-cell data of OSCC. It was found that keratinocytes closely interacted with endothelial cells, fibroblasts and macrophages through specific ligand–receptor pairs, suggesting their central regulatory role in the microenvironment of OSCC. Specifically, endothelial cell-derived SELE-CD44 signaling may promote the migration and invasion of keratinocytes, which was consistent with previous studies that CD44 (a hyaluronan receptor) enhances tumor cell metastasis through activating PI3K/AKT pathway (Zöller, 2011). APP-CD74 interactions may be involved in immune regulation, and studies have shown that CD74 plays a key role in antigen presentation and its aberrant expression may promote tumor immune escape through the NF-κB pathway (Zhou et al., 2024; Kershner et al., 2022). These indicated that the interaction network was a potential mechanism through which keratinocytes mediated the malignant progression of OSCC via immunomodulatory or paracrine signaling.

We then used the hdWGCNA technique to identify gene modules linked to keratinocyte subpopulations in OSCC, and then three hub genes (KRT16, CSTA, and CSTB) were selected to develop a diagnostic model for OSCC and validated. The expression of keratin 16 (KRT16) is closely positively related to highly differentiated OSCC (Kengkarn et al., 2020). OSCC patients with higher KRT16 expression often present poor pathologic differentiation, advanced stage, lymph node metastases, and poor survival. Inhibition of KRT16 expression could suppress the chemoresistance, invasion, migration, and metastasis of OSCC cells (Huang et al., 2019). Cysteine protease inhibitor A (CSTA) is an abundant component secreted by keratinocytes. According to in vitro simulations, human keratinocytes display impaired cell–cell adhesion in the absence of CSTA protein (Blaydon et al., 2011). Cryptogenic skin fragility is caused by a loss-of-function mutation in squamous cell carcinoma (Gupta et al., 2015). According to recent research, CSTB expression is downregulated in OSCC than in normal controls, and patients with lower expression of CSTB have worse clinicopathologic characteristics and a shorter disease-free survival. The migration, invasion, and proliferation of OSCC cells are all inhibited by overexpression of CSTB, and CSTB is associated with epidermal cell differentiation and keratinization (Xu et al., 2021). Though these results suggested that the characterized genes in keratinocytes may influence the proliferation, invasion, and metastasis of OSCC, their roles in OSCC keratinocytes and related mechanisms remained to be validated and explored by experimental studies.

The complex TME of OSCC facilitates the process of tumorigenesis and metastasis through the interaction between malignant cells and stromal cells. The TME is an ecosystem that comprises a complex system of immune cells such as T cells, dendritic cells, B cells, macrophages, and subpopulations of NK cells, which all actively participate in each stage of tumorigenesis and cancer progression (Fridman et al., 2012; Tian et al., 2017). In head and neck squamous carcinoma, stimulating local immunity may help suppress OSCC progression and enhance treatment efficacy as the head and neck region has abundant lymph nodes and blood arteries (Liang, Tao & Wang, 2020). A study reported a better prognosis for OSCC patients with higher infiltration level of T and B lymphocytes, indicating a correlation between immune cell infiltration and prognosis (Jung et al., 2020). Immature initial B cells, T cells CD8, follicular helper T cells, and Tregs are all markedly reduced in high-risk OSCC group (Liu et al., 2024). Malignant migration of OSCC cells can be promoted by the activation of M1-like tumor-associated macrophages (Xiao et al., 2018). This study found that OSCC patients had higher infiltration of cells including T cells follicular helper, M1, M0, and mast cells activated, while B cells memory, T cells CD8, and immune cells such as mast cells resting, and M2 had lower infiltration. The expression of KRT16, CSTA, and CSTB was associated with the infiltration of the vast majority of immune cells. These results suggested that the signature genes may be closely linked to immune cell infiltration in OSCC. However, the potential mechanisms of the immune landscape and characterized genes still required further exploration and validation.

The present study contained several limitations. First, the relatively small sample size in our scRNA-seq analysis may affect the generalizability of the findings, necessitating validation in larger sample size with more clinical data for confirmation. Further in vivo and in vitro studies are demanded to validate the biological functions and potential mechanisms of the three genes in OSCC keratinocytes as well as their associations with immune infiltration and chemotherapeutic agents.

Conclusion

Based on the scRNA-seq data, this study discovered KRT16, CSTA, and CSTB as the potential keratinocyte-related biomarkers for OSCC using a variety of computational techniques. These three indicators may have a significant impact on OSCC by altering keratinocyte-mediated immune cell infiltration. Further investigation of these molecular targets and their associated pathways may facilitate the development of novel targeted immunotherapy for OSCC.

Supplemental Information

Supplemental Information 1 MIQE checklist

Abbreviations

OSCC Oral squamous cell carcinoma

scRNA-seq Single-cell RNA sequencing

hdWGCNA High-dimensional weighted gene co-expression network analysis

WGCNA Weighted gene co-expression network analysis

GEO Gene expression omnibus

TME Tumor microenvironment

UMAP Uniform manifold approximation and projection

PCA Principal component analysis

KNN K nearest neighbors

DEGs Differentially expressed genes

DSigDB Drug signatures database

AUC Area under ROC curve

DCA Decision curve analysis

ROC Receiver operating characteristic

GSVA Gene set variant analysis

Additional Information and Declarations

Competing Interests

Author Contributions

Data Availability

The authors declare there are no competing interests.

Jinyu Huang conceived and designed the experiments, performed the experiments, prepared figures and/or tables, authored or reviewed drafts of the article, and approved the final draft.

Yijing Li conceived and designed the experiments, performed the experiments, prepared figures and/or tables, authored or reviewed drafts of the article, and approved the final draft.

Weiyi Lin performed the experiments, prepared figures and/or tables, and approved the final draft.

Zhiqin Wu analyzed the data, prepared figures and/or tables, and approved the final draft.

Shanshan Si conceived and designed the experiments, analyzed the data, authored or reviewed drafts of the article, and approved the final draft.

Dalong Shu conceived and designed the experiments, analyzed the data, authored or reviewed drafts of the article, and approved the final draft.

The following information was supplied regarding data availability:

Data is available at GenBank: GSE215403, GSE30784.

The raw data is available at Github and Zenodo:

https://github.com/Dalongshu994/Raw-data.

Dalongshu994. (2025). Dalongshu994/Raw-experimental-data: Raw data (rawdata). Zenodo. https://zenodo.org/records/17204553.

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
