# Peer review of "Identification of keratinocyte-associated genes for immune characterization and drug response prediction in oral squamous cell carcinoma"

_PeerJ, doi:10.7717/peerj.19953_

## Round 0.1 · original submission · Major Revisions

The two reviewers have made different degrees of revision comments on the manuscript. Please respond carefully to each one.

**Language Note:** The review process has identified that the English language must be improved. PeerJ can provide language editing services - please contact us at [email protected] for pricing (be sure to provide your manuscript number and title). Alternatively, you should make your own arrangements to improve the language quality and provide details in your response letter. – PeerJ Staff

Reviewer 1 ·

Basic reporting

no comment

Experimental design

no comment

Validity of the findings

no comment

Additional comments

In this study, multi-omics analysis revealed the molecular characteristics of keratinocytes in OSCC and proposed potential diagnostic markers. In terms of methodology, it integrates single-cell and bulk data to mine keratinocyte characteristics, and combines immunoscore to construct a prognostic model, which is innovative. However, the manuscript still needs a lot of revision.
1. Add the purpose of this paper to the Background section.
2. Please add the origin of the materials and the company names.
3. In Fig7a, the significance symbols do not correspond; some results lack significance symbols. Please add them.
4. The Methods section in the abstract is not concise. Please rewrite it.
5. The references are not up-to-date. Please add appropriate references.
6. In the Discussion section, it is suggested to add more references to research findings on keratinocytes in other cancers.
7. The English needs polishing, and the sentences are not concise and clear enough.
8. Although in vitro experiments have verified the function of KRT16, the functional validation experiments are relatively limited. Please add more experiments.
9. Further molecular biology experiments, such as Western blot and co-immunoprecipitation, could be conducted to explore the mechanisms of these genes in OSCC cells and their interactions with other signaling pathways.

Reviewer 2 ·

Basic reporting

This study identified four genes, KRT6B, KRT16, CSTB, and CSTA, as keratinocyte-related genes in OSCC through various bioinformatics techniques and constructed a nomogram with diagnostic value. Additionally, in vitro experiments showed that knocking down KRT16 significantly inhibited the viability, migration, and invasion of OSCC cells. This study provides biomarkers targeting keratinocyte-related genes for OSCC treatment, but the following shortcomings need to be addressed:
1. Although the characteristics of single-cell sequencing are described in the introduction, what are its advantages compared to previous sequencing methods?
2. I noticed that this study used the hdWGCNA method, which needs to be introduced in the introduction section, along with an explanation of the advantages of this method.
3. Please indicate whether the cells used have undergone STR authentication and whether there is mycoplasma contamination.
4. In the results section, there is no need to repeatedly describe the methods used. Please move the descriptions of the methods directly to the Materials and Methods section.
5. Why was KRT16 selected for knockout, and what is the rationale behind this choice?
6. The section "Analysis of communication between keratinocytes and other cells in OSCC" describes the receptor-ligand cell communication regulated by keratinocytes, but this is not deeply discussed in the discussion section. Further discussion is necessary.
7. Some figure captions are too concise, causing difficulties for readers. For example, in Figure 4B, the meaning of the colors and circle sizes in the figure needs to be explained.
8. It is necessary to compare the reliability of the model before and after KRT6B ablation through ablation experiments to confirm the necessity of KRT6B removal.
9. In the introduction section, it is necessary to highlight the novelty of the model constructed in this study compared to other previous OSCC models (such as PMID: 38480853, PMID: 37452322, and PMID: 35922788), emphasizing the innovative aspects of the model.
10. There are issues with the language description, as many sentences contain grammatical errors, and the formatting also needs to be adjusted according to the journal's requirements. It is recommended to have a person proficient in English and familiar with the subject matter review the language of the manuscript again, or contact professional editorial services.

Experimental design

no comment

Validity of the findings

no comment

---

## Round 0.2 · accepted · Accept

Both reviewers have recommended acceptance of your revised manuscript. I am pleased to inform you that your paper has been accepted for publication. Thank you for your careful revisions and your valuable contribution to the journal.

Reviewer 1 ·

Basic reporting

In this study, multi-omics analysis revealed the molecular characteristics of keratinocytes in OSCC and proposed potential diagnostic markers. In terms of methodology, it integrates single-cell and bulk data to mine keratinocyte characteristics, and combines immunoscore to construct a prognostic model, which is innovative. After revisions, the author provided detailed responses, and the manuscript basically met the publication standards.

Experimental design

no comment

Validity of the findings

no comment

Reviewer 2 ·

Basic reporting

no comment

Experimental design

no comment

Validity of the findings

no comment

Additional comments

This study identified four genes, KRT6B, KRT16, CSTB, and CSTA, as keratinocyte-related genes in OSCC through various bioinformatics techniques and constructed a nomogram with diagnostic value. Additionally, in vitro experiments showed that knocking down KRT16 significantly inhibited the viability, migration, and invasion of OSCC cells. This study provides biomarkers targeting keratinocyte-related genes for OSCC treatment. The author has successfully addressed the reviewers' concerns. Congratulations!